# "My friends said they should no longer be with a b*tch like me…": A qualitative study to explore the consequences of adolescent childbearing among teen mothers in Gatsibo district, Rwanda

Innocent Twagirayezu[1,2]*, Joselyne Rugema[1], Aimable Nkurunziza[1,2,3,5], Alice Nyirazigama[1,2], Vedaste Bagweneza[1], Belancille Nikuze[1], Jean Pierre Ndayisenga[1,2,4]

1 School of Nursing and Midwifery, University of Rwanda College of Medicine and Health Sciences, Kigali, Rwanda, 2 Arthur Labatt Family School of Nursing, Western University, London, Ontario, Canada, 3 Lawrence Bloomberg Faculty of Nursing, University of Toronto, Toronto, Ontario, Canada, 4 Dalla Lana School of Public Health, University of Toronto, Toronto, Ontario, Canada, 5 School of Nursing, Nipissing University, North Bay, Ontario, Canada

* twagiracent@gmail.com

## Abstract

In low and middle income countries (LMICs), the rate of teenage pregnancy continues to be high, with significant implications for maternal health. The prevalence of teenage pregnancy is worrisome in Rwanda, and little is known about the consequences faced by teen mothers aged 15–19. Therefore, the present study aims to explore the consequences of adolescent childbearing among teen mothers in Gatsibo district, Rwanda. A descriptive qualitative study design was carried out. We purposively recruited 20 teen mothers aged 15–19 from four health centers in Gatsibo district for in-depth interview. Conversations were conducted in Kinyarwanda and audio recorded. The research team, fluent in both English and Kinyarwanda, carefully listened to each recording and verbatim translated them into English ensuring they are clear and understandable. Transcripts were entered into Dedoose and then inductively and thematically analyzed. Three themes were identified: (1) self-reported health outcomes; (2) socio-economic consequences; and (3) perceived structural barriers of access to healthcare services. Teen mothers in this study reported experiencing physical and psychological problems during pregnancy and after delivery. The socio-economic consequences included lack of social support, feelings of guilt and stigmatization, discrimination, domestic violence, school interruption and dropout, and financial constraint. The structural barriers of access to healthcare included adverse health facility requirements, feeling stigmatized by adult mothers in the healthcare environment, and confronting healthcare providers' negative attitudes toward teen mothers. The results from this study indicated that adolescent childbearing is associated with several negative consequences. Therefore, there is a need in Rwanda to develop interventions addressing these consequences and revise policies and laws to ensure they support the needs of teen mothers.

**Data Availability Statement:** The data underlying the results presented in the study are available from the Figshare database: 10.6084/m9.figshare. 25715223.

**Funding:** This study was funded by The Center for International Reproductive Health Training at the University of Michigan (CIRHT-UM), USA (5000 USD to IT). The funders had no role in study design, data collection and analysis, decision to publish, or preparation of the manuscript.

**Competing interests:** The authors have declared that no competing interests exist.

## Background

Adolescent pregnancy is a worldwide public health issue that affects both industrialized and developing nations [1]. While some adolescents' pregnancies are planned and intended, the majority are not [2]. Every year, an average of 20 million adolescent girls give birth worldwide, and many of them conceive unexpectedly [3, 4]. The world's highest rate of adolescent childbearing is seen in Sub-Saharan African nations, with 21.5% occurring in eastern Africa, the region where Rwanda is located [4–6].

Early pregnancy poses substantial health risks for both the mother and unborn child [7]. Compared to older women, adolescent females in the 15–19 age range had a twofold higher risk of dying from reasons connected to pregnancy and delivery, and their children had a 50% greater chance of dying before the age of one. Additionally, adolescent mothers are more constrained in their ability to pursue educational opportunities than young women who delay childbearing [5, 6, 8, 9].

According to Gurung and colleagues [8], children born to very young mothers are more vulnerable to illnesses that affect children under the age of five, resulting in lower vaccination coverage and increased rates of malnutrition and infant mortality. This is mainly because teen mothers are vulnerable to poverty, lack knowledge regarding childcare, and often receive insufficient support from the father of the child, their parents, and the communities in which they live [5]. Furthermore, it is widely documented that teenage pregnancies have greater incidences of eclampsia, puerperal endometritis, low birth weights, and premature births than pregnancies in women aged 20–24 [5, 10]. These complications rank second among the leading causes of death associated with teenage pregnancies worldwide [11]. Stigma, family rejection, violence, and school dropout are some of the social consequences related to adolescent pregnancy [12, 13]; these factors diminish the employment opportunities available to teen mothers [4, 8].

The 2019–20 Rwanda Demographic and Health Survey reports that 6.4% of adolescent girls aged 15–19 have begun childbearing, which is higher than the overall national rate. In the same survey, Gatsibo district ranked among the top three, with an overall teenage pregnancy rate of 8.7% [14]. Despite these alarming statistics, little is known about the consequences associated with teenage pregnancies among these female adolescents. Further research is needed to better understand the full scope of the issue and inform policy decisions and interventions to support these young women. Therefore, the present study explored the consequences of adolescent childbearing among adolescent females aged 15–19 living in Gatsibo district.

## Methods

### Ethical statement

This study received approval from the Institutional Review Board of the University of Rwanda, College of Medicine and Health Sciences (Approval No:302/CMHS IRB/2022). Before data collection, all participants provided written assent, and their legal guardians signed written consent forms after being thoroughly informed about the study's purpose, their involvement, and the study's procedures. The research adhered to the principles outlined in the Declaration of Helsinki and complied with the regulations governing research involving human subjects in Rwanda.

### Study design, setting, and population

This study employed a qualitative descriptive methodology to gather rich descriptions about the consequences associated with teenage pregnancies [15]. The study participants included 20

childbearing adolescents from four different health centers in Gatsibo district, in the Eastern province of Rwanda. Researchers interviewed five participants from the neighborhood of each health center.

## Sampling and recruitment

A purposive sampling strategy was used to select adolescents who gave birth when they were between 15–19 years old [15]. After obtaining ethical approval from University of Rwanda, the research team sought permission from the mayor of Gatsibo Distinct to access the study settings. The permission letter was presented to the heads of the selected health centers, who connected researchers with community health workers (CHW). The latter identify and register pregnant women in the communities and were therefore well positioned to help recruit participants. Researchers explained the study's purpose to the CHWs and provided them with recruitment posters, which were also posted at the health centers, so that anyone who wanted to participate in the study could contact the research team to set up an appointment. Researchers included only adolescent females aged 15–19 who became pregnant while residing in the selected communities.

## Data collection instrument

In data collection, researchers used demographic questionnaire and a semi-structured interview guide developed from a thorough examination of the literature [16–19] and in consultation with two experienced researchers in the field of adolescent pregnancy. The study participants' socio-demographic characteristics, such as age, education, marital status, religion, residency, household head, parity, antenatal visit, gestational age, and occupation, were collected prior to in-depth interviews. The interview guide featured invitational questions and probing questions that encouraged participants to express all consequences related to childbearing that they identified with. The interview was conducted in Kinyarwanda because this is the only language our participants were able to express themselves in.

The interview guide commenced with general questions such as: (1) In what circumstances did you get pregnant? (2) Can you tell me how you felt after getting pregnant, during pregnancy, and after giving birth? (3) What social problems did you experience during and after your pregnancy? (4) Did you experience any obstetrical outcomes related to getting pregnant as a teen? (5) Are there any other problems that we have not discussed?

## Data collection

Participants who were invited through CHWs met one by one with two female co-investigators experienced in qualitative data collection in a clean and well-ventilated private room for an audio-recorded interview. The rooms were selected with the help of health center administration staff. Before the discussion, the co-investigators introduced themselves to the participants. They explained the purpose of the study and provided information about the risks and benefits of participating. The participants were given time to ask questions and provided with the researchers' contact information in case of any questions or concerns. Participants under the age of 18 were asked to sign an assent form and provide a signed consent form from their legal guardians. Participants aged 18 and over signed informed consent. The co-investigators explained their role in the process of data collection and set ground rules that encouraged respect for one another and informed conversation. One female co-investigator conducted the interview while another took field notes. Because the questions caused some participants to recall painful experiences related to childbearing when they were in the adolescent phase, for some, it was necessary to pause the interview. Participants were counseled by both

investigators, each of whom had mental health training. Because of the sensitivity of the topic, there was a pre-established protocol to manage emotional responses that entailed stopping the interview, counseling the participants, asking if they preferred to cease or continue the interview, and referring them to mental health professionals for further management and follow-up. Only five participants out of the twenty exhibited emotional responses such as crying and tight throat; these participants wished to continue the interview. Two follow-up calls were made after one week and one month, and the participants reported they were doing well. We reached data saturation after interviewing 20 participants because there was no new information. The shortest interview was 13 minutes and 22 seconds, and the longest was 27 minutes and 12 seconds. Daily memoing practice was done to ensure that co-investigator personal perspectives would not influence the interpretation of data. Data collection started August 30, 2022, and ended September 30, 2022.

### Data analysis

Data were translated into English and transcribed verbatim by IT, JR, AN, VB, and BN, who are fluent in Kinyarwanda. Data were organized and coded using Dedoose software. Thematic analysis was used to generate and report themes through six phases including data familiarization, generating initial codes, searching themes, reviewing themes, defining themes, and reporting themes [20]. Four research team members, including two data collectors, were involved in data analysis to ensure that the data were accurately interpreted and conclusions were based on the data collected. Before beginning to code data and search for themes, the research team familiarized themselves with the entire data set first. The team members transcribed the data themselves to become more familiar with it. This provided a useful orientation to the raw data and served as the foundation for all subsequent steps. After familiarizing themselves with the data, research team members started the initial coding by taking notes on potential data items of interest, connections, and other initial ideas using the coding template. After this step, the research team members examined the codes and extracts in order to identify broad themes. The research team held three meetings to review and define themes. All team members agreed on the final themes reported in this manuscript.

## Results

Twenty adolescents who become pregnant aged between 15–18 participated in the study. Regarding socio-economic characteristics 85% of the participants had received primary education, lived in rural areas, and were engaged in farming as their occupation. Almost all of them had experienced at least one pregnancy and had attempted to visit a clinic more than twice before giving birth (Table 1).

### Themes

Three main themes were identified during thematic analysis: (1) self-reported health outcomes; (2) socio-economic consequences; and (3) perceived structural barriers to access healthcare services (Table 2).

### Theme one: Self-reported health outcomes

Eight participants in this study reported experiencing physical and psychological problems during pregnancy and after delivery. Physical problems included prolonged labor and pain. Some participants said their skin turned yellow, which suggests anemia. Psychological

**Table 1. Demographic characteristics of the participants.**

| Age in years | Education | Marital status | Religion | Residency | Household head | Parity | Antenatal visit | Gestation weeks at delivery | Occupation |
|---|---|---|---|---|---|---|---|---|---|
| 17 | Primary education | Single | Others | Urban | Male | 1 | More than 2 | 36 | Farmer |
| 17 | Secondary | Single | Protestant | Rural | Female | 1 | More than 2 | 39 | Unemployed |
| 18 | Primary education | Single | Protestant | Urban | Female | 1 | More than 2 | 36 | Unemployed |
| 18 | No education | Single | Catholic | Rural | Female | 1 | None | 37 | Student |
| 18 | Primary education | Single | Others | Rural | Female | 1 | More than 2 | 36 | Farmer |
| 17 | Primary education | Single | Protestant | Rural | Male | 1 | Between 1 and 2 | 36 | Unemployed |
| 17 | Primary education | Single | Protestant | Rural | Male | 0 | Between 1 and 2 | 37 | Farmer |
| 15 | No education | Single | Catholic | Rural | Female | 1 | Between 1 and 2 | 39 | Farmer |
| 16 | Primary education | Single | Protestant | Rural | Female | 1 | Between 1 and 2 | 39 | Student |
| 17 | Primary education | Single | Others | Rural | Male | 1 | Between 1 and 2 | 36 | Farmer |
| 18 | Primary education | Single | Catholic | Rural | Female | 1 | More than 2 | 40 | Farmer |
| 17 | Primary education | Single | Protestant | Rural | Female | 1 | More than 2 | 40 | Farmer |
| 17 | No education | Single | Catholic | Rural | Female | 1 | More than 2 | 39 | Farmer |
| 18 | Primary education | Single | Catholic | Rural | Female | 1 | More than 2 | 40 | Farmer |
| 15 | Primary education | Single | Protestant | Rural | Male | 1 | More than 2 | 39 | Farmer |
| 18 | Secondary education | Single | Protestant | Rural | Female | 1 | More than 2 | 39 | Farmer |
| 17 | Secondary education | Single | Catholic | Rural | Female | 1 | More than 2 | 39 | Farmer |
| 17 | Secondary education | Single | Protestant | Rural | Male | 1 | Between 1 and 2 | 30 | Farmer |
| 15 | Primary education | Single | Protestant | Rural | Female | 1 | More than 2 | 40 | Unemployed |
| 17 | Secondary education | Single | Protestant | Urban | Female | 1 | More than 2 | 40 | Farmer |

**Table 2. Themes and sub-themes.**

| Themes | Sub-themes |
|---|---|
| **Self-reported health outcomes** | Physical |
| | Psychological |
| **Socio-economic consequences** | Loss of future |
| | Lack of social support |
| | Stigma and discrimination |
| | Domestic violence |
| | School interruptions and dropout |
| | Financial constraints |
| **Perceived structural barriers to access healthcare services** | Health facility requirements |
| | Judgment from adult mothers in the healthcare environment |
| | Healthcare providers' negative attitudes toward teen mothers |

problems included fear of disclosing the pregnancy, intense stress, anxiety, depression, feelings of guilt, and poor coping mechanisms such as sleeplessness and suicidal thoughts.

A few of the participants had premature deliveries, and others underwent cesarean section. Some participants reported experiencing anemia during pregnancy, and three reported high blood pressure.

For instance, one teen mother noted, "*Throughout my pregnancy, I was weak, my skin was always yellow and that left me always feeling worn out.*" Site 4, P2.

Another participant added, "*I was referred immediately from the health center to the district hospital because my blood pressure was high and told that it would be severe.*" Site 4, P5.

Thirteen participants experienced mental problems such as difficulty accepting that they were pregnant, fear of disclosing the pregnancy, and intense stress, anxiety, and depression related to the delivery process and altered relationships with their family members when they learned of the pregnancy.

One participant explained, "*After learning that I was pregnant, I felt overwhelmed with despair but eventually accepted the reality of the situation.*" Site 4, P5.

Other participants expressed that, even though they had accepted giving birth, they experienced fear and intense stress related to pregnancy and the delivery process, as noted in one quote: "*The fear I experienced was specifically related to the delivery process and unprepared parenthood.*" Site 1, P1.

Another participant added, "*I was filled with fear and apprehension regarding my father's reaction upon learning about my pregnancy.*" Site 3, P2.

Most of the study participants narrated that after learning that they were pregnant, they felt guilty and ashamed. For instance, one study participant explained, "*I deeply regretted getting pregnant at my age, as it was an unplanned occurrence. It was a shame on my family.*" Site 1, P3.

Five participants reported that these psychological problems led to poor coping mechanisms. Some experienced trouble sleeping. As stated by one participant, "*I felt depressed when I called the father of this child and found that the phone was off. After being discharged from hospital, I found it [phone] inaccessible. I could not sleep for a long time.*" Site 1, P1.

A few other participants reported having suicidal thoughts, and some attempted to commit suicide, as reflected in the following quote:

"*I attempted suicide using a lace, but during the act, I felt physical pain in my body, including headaches and earaches. The lace broke and fell off. On another occasion, I tried to end my life by going to Lake Muhazi, which was far from my home. However, while on my way, it started raining heavily, and I thought about my child being left alone at home. I quickly turned back and rushed home, where I found my child crying. Seeing my child in that state, I was overwhelmed with emotion, sought his or her forgiveness, and made a commitment to never engage in such actions again.*" Site 4, P5

A few other participants reported that due to the lack of social support and being ostracized by their families, they did not want to raise their babies. They felt abortion was a solution. For instance, one participant explained:

*Given the circumstances I was facing, my family decided to disown me, leaving me homeless. Throughout that difficult period, the idea of having an abortion crossed my mind. However, my conscience strongly opposed the notion. I struggled with these conflicting thoughts until I ultimately made the decision to give birth.* Site 1, P2

The study participants reported that teenage pregnancy was associated with a variety of health concerns, such as mental health problems and poor physical health. This is likely due to the fact that teenage mothers often lack the support and resources needed to take care of their health.

### Theme two: Socio-economic consequences

The participants in this study reported various socio-economic consequences of teenage pregnancy at individual and societal levels. These consequences included loss of future, lack of social support, stigma and discrimination, family conflict, school interruptions and dropout, and financial constraints.

Fifteen study participants narrated that after learning they were pregnant, they felt disappointed in their families, and others felt their dreams had shattered.

For instance, one study participant explained, *"It was a very difficult time for me as I was worried about the future of my child. . . I couldn't help but think about my own life. . . .. It was also hard for me to see my colleagues going back to school while I had to put my education on hold due to the pregnancy."* Site 3, P2.

After getting pregnant or after delivery, teen mothers in this study reported lacking support from their partners and families. Most study participants reported being abandoned by their partners.

For instance, one study participant explained, *"After engaging in sexual intercourse with my boyfriend and revealing my pregnancy, he informed me that I should have an abortion as he wasn't ready for marriage. Thus, he did not support me."* Site 2, P3.

Another participant confirmed, *"I was left alone by the man who had impregnated me, without any support or even providing the necessary materials as the delivery approached. This caused me a lot of stress and frustration, especially when I found out that he had gotten married to another woman."* Site 2, P2.

Teen mothers reported that their pregnancies led to family conflicts such as quarrels, misunderstandings between parents, and family separation. Family conflicts were found to be the primary source of psychological problems experienced by teen mothers after becoming pregnant. Some families are never able accept their child's pregnancy, even rejecting their grandchild. On the other hand, teen mothers expressed that family support, when present, was the pillar of their wellbeing and psychological stability.

*He [my father] was very harsh to me, to the extent that I lost my appetite due to his words. Even before, when he used to drink, he would treat me harshly by claiming that he didn't give birth to a child like me and would speak unfriendly words to my mother, accusing her of my problems. It became difficult for me to eat in such a situation.* Site 2, P1

*Because my family didn't accept my situation of being pregnant, it was a big challenge because no one cared about me. Even when I need something, they said to go and seek assistance from the [person] responsible for the pregnancy, while I was not aware of where he was.* Site 1, P5

Some study participants revealed that they experienced stigmatization and discrimination from their peers and community because of their pregnancy. Among 20 teen mothers interviewed, 15 of them described becoming estranged from their peers, childhood friends, and schoolmates. Some of these peers now labeled them as prostitutes or misbehaved girls; they talked behind their backs saying they cannot give good advice because they got pregnant early.

For example, one participant noted, *"My friends said they should no longer be with a bitch like me. They claimed that I could influence others negatively."* Site 1, P3.

Another participant added, *"They broke up our relationship and shit me claiming that I am no longer a girl but [a] woman."* Site 4, P2.

All 20 interviewed teen mothers dropped out of school. Many became pregnant while they were still studying; however, a few became pregnant after they had dropped out due to other reasons. Only one teen mother wished to get back to school.

For instance, one study participant highlighted, *"After getting pregnant, I became less interested in school activities. Finally, I dropped out."* Site 1, P4.

Lastly, teen mothers in this study reported some financial constraints as a result of getting pregnant. These long-term economic consequences were due to lack of employment opportunities, difficulty maintaining stable jobs, and childcare responsibilities, as reflected in the following quote:

> *You could have money and wish to buy something, but inevitably, you think about your child, my child lacks shoes, if I could buy them, how would I go to vaccination appointments. . ., I am going there without taking porridge or without shawl of my baby? without a baby umbrella then you think deeply about your life situation and realize that the reality of life changes.* Site 3, P5

This was also confirmed by another participant who explained, *"Having a child made it really difficult for me to complete my education and enroll in college. Finding a job without a diploma has proven difficult for me."* Site 4, P2.

Others reported difficulty finding a job due to their pregnancy status, as one participant noted, *". . . No one can give you a job when you are pregnant because they fear that you might fall and get injured. As a result, you can't find money needed to buy things like baby clothes."* Site 1, P3.

Understandably, adolescents in this study faced several socio-economic consequences resulting from the loss of education and employment opportunities. These consequences included poverty, family instability, and difficulty accessing resources. This lack of opportunities profoundly impacted the adolescents' ability to plan for the future, realize their dreams, develop self-efficacy, and gain access to the resources needed to succeed.

### Theme three: Perceived structural barriers to accessing healthcare services

All study participants pointed out barriers faced by teen mothers when accessing health services. These barriers included adverse health facility requirements, being stigmatized by adult mothers in the healthcare environment, and healthcare providers' negative attitudes toward teen mothers.

Teen mothers were challenged by health center policies and procedures concerning the first antenatal care visit, where they were required to be accompanied by their partner. Most of the teen mothers had conflicts with their partners, many of whom refused the pregnancy. Only one out of 20 interviewed teen mothers came with her partner for the first visit, and this participant was later abandoned by her partner.

> *They required that I bring my husband and explained to them that I was single and not married, they asked me to bring paper from the local leader proving that I don't have a husband. After getting the required paper, they assessed me again and recommended that I have to bring my husband next time. I communicated with him but he declined. When I went back and said that he refused, they reassessed and registered me.* Site 3, P2

Some study participants reported being asked about community health insurance while they could not afford to pay for it.

For instance, one participant reported, *"You know, it's very known that you can't be received without a health insurance and I didn't have it. So, I had to go back to the community health worker."* Site 2, P4.

Fear of being stigmatized by other adult mothers at the healthcare center was also reported by some study participants:

*"I was not comfortable going to the clinic because it happened once other mothers asked me many questions. For example, one mother asked me how I got pregnant at a young age. I felt embarrassed and frustrated."* Site 3, P3.

These participants added that they were not comfortable going to antenatal care since they would meet with their neighbors and family friends, whom they did not want to encounter while they were pregnant.

A few other participants reported the healthcare providers' negative attitudes toward teen mothers and described experiencing rude behavior and reproach.

For example, one study participant explained, *"A nurse was giving me instructions and I could not follow them well, and she rudely said, 'It's not surprising that you are pregnant because you don't listen.'"* Site 1, P1.

Another participant added, *"A nurse looked at me for a few seconds and asked me: How could you do this at this age? This is inappropriate at your age".* Site 2, P2.

This illustrates that pregnant adolescents who accessed health services faced many challenges related to health center guidelines and procedures. These challenges included increased community stigma, even in the healthcare environment, and healthcare providers' negative attitudes toward teen mothers. These challenges often cause pregnant adolescents to avoid seeking health services, leading to poor health outcomes for both mother and child.

## Discussion

This study aimed to explore and describe the consequences of adolescent childbearing in Rwanda. From the participants' narratives, we identified health outcomes, socio-economic consequences, and barriers of access to healthcare services.

Our findings revealed that teen mothers faced health problems, both physical and psychological, as a result of teenage pregnancy. Physical outcomes include signs and symptoms of anemia, high blood pressure, and prolonged labor. Adolescent mothers (aged 10–19) are more likely to develop eclampsia, puerperal endometritis, and systemic infections, and their babies are more likely to have low birth weights, preterm births, and severe birth defects [21]. These outcomes highlight the importance of comprehensive reproductive health education and healthcare access for teenagers.

In our study, some participants expressed that they experienced mental health problems, including difficulty accepting they were pregnant, fear of disclosure, intense stress, anxiety, and depression. These findings are consistent with a systematic review that has been conducted in Sub-Saharan African countries to assess the mental health problems and gaps in service provision experienced by pregnant teens and young women [22]. In this review, depression was the most studied mental health problem followed by substance use, stress, and loss of self-esteem. In Rwanda, a recent study found that 48% of teen mothers develop postpartum depression [23]. In the Rwandan context, teen mothers develop mental health problems because they are less able to talk to people about their challenges [24]. However, data in

developed countries show that the rate of mental health problems is reportedly low among teen mothers [25], reflecting the available resources to support them. This implies that interventions are needed to prevent the high rates of depression among teenage mothers in low- and middle-income countries (LMICs).

In our study, a few teen mothers reported suicidal thoughts, and others tried to commit suicide as a result of the challenges faced after getting pregnant. Suicidal attempts are common in African settings [22]. A variety of factors contribute to suicide among teen mothers, including lack of social support and negative feelings about their personal health status [26, 27]. Teen mothers and young women in sub-Saharan Africa might also be more likely to engage in self-harm and suicidal behavior due to the sociocultural values placed on them [22]. These risk factors should be considered when providing services to teen mothers.

Because of the challenges they faced, teen mothers in this study viewed abortion as a potential solution. This finding is consistent with results from another study in the Rwandan context, which identified abortion as a potential outcome among teen mothers due to paternity denial, the possibility of assuming parenting responsibility alone and potential challenges having a child at a young age may bring to their future marriage [24]. Data from 2019 indicate that over half of unintended pregnancies among adolescent girls aged 15–19 result in abortions, usually performed in unsafe conditions in LMICs [28]. Our study did not focus on who attempted to abort or had aborted their pregnancies. Therefore, future studies should explore abortion as one of the potential pregnancy resolutions among teen mothers and how interventions can be developed to prevent unsafe abortions.

In the current study, some teen mothers reported several socio-economic consequences as a result of teenage pregnancy. Some mentioned that they do not receive support from families and partners, which exacerbates their financial situations. These findings are consistent with another study that has been conducted in Kenya [29]. This study found that parents of the teen mother or of the biological father typically provide negligible support. Despite their sympathy for their daughters' pregnancies, the mothers could not support them due to food shortages and lack of resources. Developing and implementing strategies to support adolescent mothers, especially those with limited social support, is an urgent need.

Perceived stigma and discrimination were also reported by most teen mothers in this study. They reported diminished social contexts and fear of being seen outside their home places. Others reported a break in relationships with their close friends. Still others reported conflicts with their parents, especially their fathers. Social stigma attached to teenage pregnancy has been reported in Rwanda and other African settings [24, 29, 30]. Culturally, in most countries, it is highly stigmatized to engage in premarital sexual activity, and pregnancy disclosure implies concurrent sexual activity, whether it was consented to or not [24, 31].

Teen mothers described experiencing problems in their family relationships, such as quarrels, misunderstandings between parents, and family separation. Family conflicts were found to be the primary source of psychological problems faced by teen mothers after becoming pregnant. The announcement of pregnancy by a teen can cause arguments, quarrels, and strained relations between the teen and her parents. These young women may even leave their homes due to fear of physical aggression, verbal assault, and the potential imposition of abortion [32]. Support networks can greatly help teen mothers navigate family conflicts. Obtaining guidance, emotional support, and practical resources from peers, mentors, healthcare providers and community organizations can help them communicate effectively with their parents and find alternatives when deciding to leave home. A support network can also offer teen mothers a safe place to share their experiences, ask for advice, and learn from others who have experienced similar struggles.

While all governments are required by international law to provide an education for all children, some African countries currently prohibit pregnant teens and mothers from attending school [33]. In Rwanda, pregnant teens and teen mothers are allowed to attend schools; however, teen mothers in this study reported experiencing some school interruptions and dropouts. The prevalence of school dropout among teen mothers continues to be worrisome in African countries [34, 35]. It is necessary to tailor programs specific to the needs of pregnant teens who drop out of school, teens in school who have become pregnant, and teens who are at risk of becoming pregnant. Some of the documented strategies include: comprehensive sexual education, establish support system and accommodation in school, enabling free access to health care services and contraceptives [6, 36].

Teen mothers in this study reported struggling to care for themselves and their children. Many faced economic insecurity because they received little or no support from their families. These findings were also reported in other countries and in Rwanda [24, 37]. Moreover, impregnating a teen in Rwanda is illegal whether a child consents or not [38]. Therefore, the perpetrators often choose to abandon the teen mother and sometimes flee the country [39], which is a financial burden for the teen mother because she cannot receive support from the partner. Policies should be revised to ensure boys and men who impregnate young girls are involved in supporting adolescent mothers and their children. Involving boys and men in supporting adolescent mothers can help break the cycle of teenage pregnancy by promoting responsible fatherhood and positive male role models. In addition, it can provide emotional and financial support to young mothers, leading to better outcomes for both mother and child.

Some study participants reported barriers to accessing healthcare services, such as being asked to bring their partners to the health facility at the first antenatal care visit. These findings were also reported in previous studies [40–42]. Another issue reported was the need to present community health insurance which they could not afford. Some teen mothers explained that they were from low-income families that could not afford to pay 3,000 Rwandan francs, which is the per person cost of a health insurance policy in Rwanda. Moreover, to be given health insurance, all family members need to pay [43], which is an added barrier for teen mothers who are ostracized by their families. The community health insurance policy should consider the particularities of teen mothers to ensure they can access services.

In this study, teen mothers shunned health services due to fear of being stigmatized by other adult mothers in the healthcare setting. Teen mothers were accused of violating the age norms of parenting. These findings imply that to ensure privacy, teen mothers should not share the waiting room and antenatal care with adult mothers. A few other participants reported the healthcare providers' negative attitudes toward teen mothers, citing examples such as rude behavior, scolding, and being blamed for getting pregnant at an early age. These findings corroborate other studies that were conducted in different settings [40, 44, 45]. The results suggest that healthcare providers should be aware of their potential biases against teen mothers and provide better support for them to ensure a positive and safe experience.

## Limitations

This study has some limitations. First, it was conducted in one district, so the findings may not be generalized to all teen mothers in the whole country. Second, the sample size included teen mothers from low-income families without social support. Participants with social support or from higher-income families might have reported different experiences. Third, almost all study participants lived in rural areas. This may have impacted the study results as rural areas feature different resources and support systems compared to urban areas. Finally, the study only included teen mothers with primary education, so the findings may not be applicable to

those with secondary education. Future research should employ a mixed methods approach to better capture the consequences of teen pregnancy for mothers from differing educational backgrounds. Additionally, research should investigate the impact of social support and income on the consequences associated with teenage pregnancy in different communities. Finally, research should include more diverse participants such as health care provider and parents to better understand the topic.

## Conclusion

This study revealed that being a teenage mother in the Rwandan context is associated with several challenges, including poor health outcomes, socio-economic consequences, and structural barriers to accessing healthcare services. Governments and organizations should develop and implement gender-sensitive policies and programs to support adolescent mothers. To prevent teenage pregnancies, different stakeholders should create awareness about the consequences of teenage pregnancies and empower both girls and boys with the necessary knowledge and skills before they reach the adolescent stage. To prevent physical and mental health problems, teenage mothers need sufficient access to healthcare services. This includes providing teenage mothers with adequate health information and access to community resources and support services. To cope with the socio-economic challenges of becoming a young parent, teenage mothers should be provided access to economic and social services, especially those from rural and low-income families. Lastly, policies and laws should be revised to address structural barriers to healthcare services. This includes curtailing the mandated presence of their partners, providing teenage mothers with free health insurance, establishing private antenatal areas, and offering training to healthcare providers on how to dismantle their negative attitudes toward adolescent mothers.

## Acknowledgments

We acknowledge the 20 teen mothers for participating in this study.

## Author Contributions

**Conceptualization:** Innocent Twagirayezu, Joselyne Rugema, Alice Nyirazigama, Vedaste Bagweneza, Belancille Nikuze.

**Data curation:** Innocent Twagirayezu, Joselyne Rugema, Alice Nyirazigama, Vedaste Bagweneza, Belancille Nikuze.

**Formal analysis:** Innocent Twagirayezu, Joselyne Rugema, Aimable Nkurunziza.

**Funding acquisition:** Innocent Twagirayezu.

**Investigation:** Innocent Twagirayezu.

**Methodology:** Innocent Twagirayezu, Aimable Nkurunziza, Alice Nyirazigama.

**Project administration:** Innocent Twagirayezu, Joselyne Rugema.

**Resources:** Innocent Twagirayezu.

**Software:** Innocent Twagirayezu, Aimable Nkurunziza, Jean Pierre Ndayisenga.

**Supervision:** Innocent Twagirayezu.

**Validation:** Innocent Twagirayezu, Aimable Nkurunziza, Jean Pierre Ndayisenga.

**Visualization:** Innocent Twagirayezu, Aimable Nkurunziza, Jean Pierre Ndayisenga.

**Writing – original draft:** Innocent Twagirayezu, Joselyne Rugema, Aimable Nkurunziza.

**Writing – review & editing:** Innocent Twagirayezu, Joselyne Rugema, Aimable Nkurunziza, Alice Nyirazigama, Vedaste Bagweneza, Jean Pierre Ndayisenga.

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
