## [Decision Letter · Decision Letter 0]

29 Mar 2024

PGPH-D-23-02505

“…My friends said they should no longer be with a b*tch like me…”: A qualitative study to explore the consequences of adolescent childbearing among teen mothers in Gatsibo district, Rwanda

Dear Dr. Twagirayezu,

Thank you for submitting your manuscript to PLOS Global Public Health. After careful consideration, we feel that it has merit but does not fully meet PLOS Global Public Health’s publication criteria as it currently stands. Therefore, we invite you to submit a revised version of the manuscript that addresses the points raised during the review process.

*Comments from PLOS Editorial Office: We note that one or more reviewers has recommended that you cite specific previously published works. As always, we recommend that you please review and evaluate the requested works to determine whether they are relevant and should be cited. It is not a requirement to cite these works. We appreciate your attention to this request.*

We look forward to receiving your revised manuscript.

Kind regards,

Dandara de Oliveira Ramos, PhD

Academic Editor

Journal Requirements:

Additional Editor Comments (if provided):

Reviewers' comments:

Reviewer's Responses to Questions

**Comments to the Author**

1. Does this manuscript meet PLOS Global Public Health’s publication criteria? Is the manuscript technically sound, and do the data support the conclusions? The manuscript must describe methodologically and ethically rigorous research with conclusions that are appropriately drawn based on the data presented.

Reviewer #1: Yes

Reviewer #2: Yes

2. Has the statistical analysis been performed appropriately and rigorously?

Reviewer #1: N/A

Reviewer #2: N/A

3. Have the authors made all data underlying the findings in their manuscript fully available (please refer to the Data Availability Statement at the start of the manuscript PDF file)?

Reviewer #1: Yes

Reviewer #2: Yes

4. Is the manuscript presented in an intelligible fashion and written in standard English?

Reviewer #1: No

Reviewer #2: No

5. Review Comments to the Author

Reviewer #1: Whilst this paper addresses an important and relevant topic, there are numerous errors and typos, which detract from the content. I believe that the topic and the data are important and should be published, however in order to get it to a publishable standard, the paper would need a thorough copy edit. In the specific comments below I picked up some mistakes and typos, but I am sure there are some I missed. An English language editor should do a detailed edit to ensure these are captured.

The authors make some important statements about the mental health outcomes related to teenage pregnancy, which are crucial to address. It is critical to consider how many of these aspects interact and intersect and in doing so enhance adolescent girls and young women’s (AGYW) risk of negative physical and mental health outcomes. Literature shows that the stress related to the discovery of an unexpected pregnancy is compounded by the shame and social stigmatisation of teenage pregnancy, and the ensuring social isolation from family and community increases the risk for psychological distress amongst teenagers.

The stigmatisation of teen pregnancy is a critical factor. The framing of pregnancy during adolescence as a social problem means that pregnant teens receive limited social support, which in turn is linked to poor mental health outcomes. Emotional isolation and lack of support, especially when faced with stressors such as the discovery of an unexpected pregnancy, negatively impacts mental health. The emotionally distressing aspects of unexpected discovery of pregnancy, combined with a lack of social support, contribute to the high rates of depression amongst AGYW. Additionally, an ‘unintended’ pregnancy can compound pre-existing social and economic vulnerabilities, and result in heightened feelings of stress and unhappiness. Adolescent pregnancy poses a significant mental health burden, predisposing AGYW to adverse mental health outcomes, with depression and anxiety being the most common. In resource-deprived settings in sub-Saharan Africa, pregnancy amongst AGYW is associated with adverse mental health outcomes and psychosocial stresses including stigma and discrimination. Pregnancy may exacerbate existing social and contextual stressors, adding additional stressors such as interpersonal relationship challenges, regret around ‘unintended’ pregnancies, and depression. Lastly, violence in relationships, a lack of emotional support from family and partners, and financial insecurity interact to exacerbate AGYW vulnerability to poor mental health and SRH outcomes.

Specific comments

Line 37: “descriptive” doesn’t need to be capitalised

Line 38: “for interviewing”? –revise sentence

Line 39: “conversations were conducted in Kinyarwanda and audio recorded”

Line 39: translated inro English transcripts? What was the translation process – provide more detail

Line 52: did the study examine consequences?

Line 85-87: “Despite these alarming statistics, little is known about the consequences associated with teenage pregnancies among these female adolescents to understand the full scope of the issue better.” – something missing from this sentence? Needs to be revised

Line 95: capitalise “Eastern” if pronoun

Line 109: who is “us”? – preferable to write in third person – i.e. “the research team” – same comment with “we” in rest of paragraph

Line 144: all interviews were conducted in a 2 day period??

Line 160: where does the data in the table come from? Was a demographic questionnaire also administered? If so, describe it in methods

Line 171: skin turning yellow meaning anaemia or jaundice?

Line 218: what is meant by “loss of future”?

Line 222: what is meant by “disappointed in their families” ?

Lines 234-240: sentences need revision – do not make sense and require punctuation

For example Line 234: “their pregnancy is essence”?

Line 237: “most the teen fathers are the first to traumatize their daughters but fewest mothers also have been found” ?? – does not make sense

Line 251: refrain from using the term ‘prostitutes’ unless it is a direct quotation indicated by quotation marks. The term sex worker is more appropriate

Lines 251-252: “misbehaved girls, they talk on their back saying none can give good advices as they got pregnant early” – does not make sense

Line 271: “the hiring people” – do you mean employers?

Line 287: “and the guy escaped the area” ??

Line 293: “Some study participants reported being asked a community health insurance while they could not afford it”- word missing? ?

Line 343: “they faced, they felt” – cut the 2nd they - revise

Line 366: “voiced that they encountered” – revise

Suggested additional reading:

Choi KW, Smit JA, Coleman JN, Mosery N, Bangsberg DR, Safren SA, et al. Mapping a syndemic of psychosocial risks during pregnancy using network analysis. Int J Behav Med. 2019;26:207–16.

Duby, Z., McClinton Appollis, T., Jonas, K. et al. (2020). “As a Young Pregnant Girl… The Challenges You Face”: Exploring the Intersection Between Mental Health and Sexual and Reproductive Health Amongst Adolescent Girls and Young Women in South Africa. AIDS and Behavior, 25(2), 344-353. https://doi.org/10.1007/s10461-020-02974-3

Osok J, Kigamwa P, Vander Stoep A, Huang K-Y, Kumar M. Depression and its psychosocial risk factors in pregnant Kenyan adolescents: a cross- sectional study in a community health Centre of Nairobi. BMC Psychiatry. 2018;18:136.

Watt MH, Eaton LA, Choi KW, Velloza J, Kalichman SC, Skinner D, et al. “It’s better for me to drink, at least the stress is going away”: perspectives on alcohol use during pregnancy among South African women attending drinking establishments. Soc Sci Med. 2014;116:119–25.

Reviewer #2: This manuscript qualitatively examines the consequences of adolescent pregnancy in Gatsibo district in Rwanda. The authors find that adolescent mothers face physical and psychological problems, stigma and barriers to accessing healthcare services.

The paper is well structured and covers the most important aspects of their data and published research that corroborates their findings.

However some finer points remain.

1. There are several grammatical and punctuation mistakes. I would recommend authors to have their manuscript proof-read by a native speaker or use online services to rectify these.

2. Some parts of the abstract and introduction need better context setting in terms of regions the authors want to focus on. Please see the comments in the attached file.

3. The method section needs some clarifications and more details on the data collection for example the positionality of the authors in comparison to the participants. Please see the comments in the attached file.

4. In the results section, please follow the same structure with an explanation of the main themes then evidence in form of quotes. Also add the number of participants who reported a particular consequence. Some paragraphs have these counts and others do not.

5. The discussion is well written but need some more reflection on topics like role of men, partners, boys in the broader scope of intervention design and supporting girls. The topic of marital status should also be discussed within the SSA context and in other LMICs. For example, adolescent pregnancy happens within the context of marriage in south asia while this is not the case in this study.

6. a section recommendations on how to supprt girls is missing. Also comment on the widespread application of gender transformative interventions or comprehensive sexual health education and how they can potentially support girls in this context.

6. PLOS authors have the option to publish the peer review history of their article (what does this mean?). If published, this will include your full peer review and any attached files.

**Do you want your identity to be public for this peer review?** For information about this choice, including consent withdrawal, please see our Privacy Policy.

Reviewer #1: No

Reviewer #2: **Yes: **Shruti Shukla

---

## [Decision Letter · Decision Letter 1]

26 Aug 2024

“…My friends said they should no longer be with a b*tch like me…”: A qualitative study to explore the consequences of adolescent childbearing among teen mothers in Gatsibo district, Rwanda

PGPH-D-23-02505R1

Dear Mr. Twagirayezu,

We are pleased to inform you that your manuscript '“…My friends said they should no longer be with a b*tch like me…”: A qualitative study to explore the consequences of adolescent childbearing among teen mothers in Gatsibo district, Rwanda' has been provisionally accepted for publication in PLOS Global Public Health.

Best regards,

Julia Robinson

Executive Editor

Reviewer Comments (if any, and for reference):

Reviewer's Responses to Questions

**Comments to the Author**

1. If the authors have adequately addressed your comments raised in a previous round of review and you feel that this manuscript is now acceptable for publication, you may indicate that here to bypass the “Comments to the Author” section, enter your conflict of interest statement in the “Confidential to Editor” section, and submit your "Accept" recommendation.

Reviewer #1: All comments have been addressed

2. Does this manuscript meet PLOS Global Public Health’s publication criteria? Is the manuscript technically sound, and do the data support the conclusions? The manuscript must describe methodologically and ethically rigorous research with conclusions that are appropriately drawn based on the data presented.

Reviewer #1: Yes

3. Has the statistical analysis been performed appropriately and rigorously?

Reviewer #1: N/A

4. Have the authors made all data underlying the findings in their manuscript fully available (please refer to the Data Availability Statement at the start of the manuscript PDF file)?

Reviewer #1: Yes

5. Is the manuscript presented in an intelligible fashion and written in standard English?

Reviewer #1: Yes

6. Review Comments to the Author

Reviewer #1: Authors have addressed comments satisfactorily.

7. PLOS authors have the option to publish the peer review history of their article (what does this mean?). If published, this will include your full peer review and any attached files.

**Do you want your identity to be public for this peer review?** For information about this choice, including consent withdrawal, please see our Privacy Policy.

Reviewer #1: No
